# Direct-Grown Helical-Shaped Tungsten-Oxide-Based Devices with Reconfigurable Selectivity for Memory Applications

Ying-Chen Chen [1],*, Yifu Huang [2], Sumant Sarkar [3], John Gibbs [4] and Jack Lee [2]

1 Department of Electrical and Computer Engineering, Northern Arizona University, Flagstaff, AZ 86011, USA
2 Department of Electrical and Computer Engineering, The University of Texas at Austin, Austin, TX 78712, USA
3 Lam Research, Tualatin, OR 94538, USA
4 Department of Applied Physics and Materials Science, Northern Arizona University, Flagstaff, AZ 86011, USA
* Correspondence: ying-chen.chen@nau.edu

**Abstract:** In this study, a direct-grown helical-shaped tungsten-oxide-based (h-WO$_x$) selection device is presented for emerging memory applications. The selectivity in the selection devices is from 10 to $10^3$ with a low off-current of 0.1 to 0.01 nA. In addition, the selectivity of volatile switching in the h-WO$_x$ selection devices is reconfigurable with a pseudo RESET process on the one-time negative voltage operations. The helical-shaped selection devices with the glancing angle deposition (GLAD) method show good compatibility, low power consumption, good selectivity, and good reconfigurability for next-generation memory applications.

**Keywords:** helices; memristor; RRAM; selector; volatile switching

## 1. Introduction

With high-demand computing, the emerging memory with high-density storage and computational functions attracts attention. An inevitable issue needs to be solved before enabling the high-density crossbar array, called "sneak path currents (SPC)", i.e., the crosstalk effect resulting from the leakage current contributed by the neighboring memory cells subverting the development of crossbar arrays [1,2]. Currently, the configuration of one transistor–one memory (1T-1R) is utilized to solve the SPC issue, which relies on the rectifying behavior of the three-terminal transistor to suppress the SPC and mitigate the programming errors in the memory array [3–6]. However, this approach with a three-terminal transistor integration increases the cost and fabrication complexity while limiting the scalability per bit and the whole memory array. The self-rectified memory and good performance selection device are in pressing need of development [6–8].

Recently, self-rectified memory applications without the integration of the selection devices were published. However, the filamentary-based memory has an intrinsic high off-current (~$10^{-6}$) after electroforming, which leads to the bottleneck of energy efficiency as well as noise immunity in the current self-rectified memory [9–12]. To mitigate the development gap between 1T-1R and self-rectified memory, the novel two-terminal selection device is proposed and demonstrated with simplified fabrication, low operation energy, and scalability. To investigate the system level's latency and writing energy, SPICE modeling on 1T-1R, dynamic random-access memory (DRAM), and resistive random-access memory (RRAM) (~1 Gb) have been reported [13]. The 2T-1R and 3T-1R configurations are also published to mitigate the SPC-induced operation errors while reducing the energy consumption [14–17]. Despite there are breakthroughs in circuit- and system-level studies, the device-level RRAM characteristics still lack investigation. To overcome the limitations on device-level memory development, the power consumption and SPC-induced errors in the array operation need an urgent solution. In this study, the direct-grown two-terminal selection device that uses the nano helical structures is presented with reconfigurable

selectivity. The results present the reconfigurable volatile switching (VS) behaviors of WO$_x$ helical-shaped films, which is suggested as the solution for the SPC issue without compromising the structure simplicity, which enables the low power, high density, CMOS compatible technology towards the crossbar memory array applications.

## 2. Results and Discussion

The fabrication process is shown in Figure 1a. After the wafer cleaning process, 100 nm of gold (Au) was deposited as the bottom electrode (BE) using a physical vapor deposition (PVD) method while keeping the substrate at an angle of 0 degree. The WO$_x$ helices were grown using the glancing angle deposition (GLAD) method (Figure 1b) [18]. Briefly, the WO$_x$ source material (WO$_x$ pellet, Kurt J. Lesker) was kept in a Mo crucible, and e-beam deposition method was used to deposit the material on the substrate, while the substrate was rotated. Furthermore, the substrate was kept at an angle of 86 degrees with respect to the vapor plume, to achieve the shadow effect needed by GLAD to create helices. The helical-shaped thin film, i.e., insulator layers has been deposited by the e-beam evaporation including helical WO$_x$ (h-WO$_x$). The total helical wire length was designed to be modified from 50 nm, 100 nm, 200 nm (approximately 1 full turn, and 4 turns of helix). The height of the helical structure is examined to be approximately 60 to 80% of the total wire length [19]. The growth of the helices occurred at a rate of around 0.7 Å/s. This was monitored using an in-vacuo quartz-crystal microbalance (Inficon, Bad Ragaz, Switzerland). Then, 5 nm of SiO$_x$ has been deposited by atomic layer deposition (ALD) for isolating the top electrode from the bottom electrode and preventing the shortened circuit. The devices without the SiO$_x$ capping layer are fabricated as references. The Keysight B1500 with EPS probe station is utilized for electrical characterization and analysis.

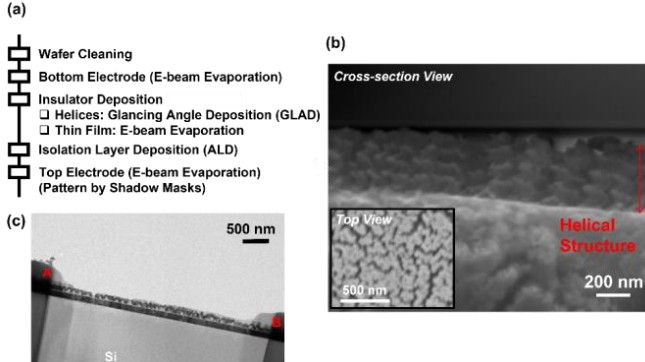

**Figure 1.** (**a**) Fabrication process for helical-shaped nanostructure, (**b**) cross-section image of the helical structure by SEM image (inset: top view of as-deposited helical structures), (**c**) helical structure deposited and fill in the via opening by shadow masks.

The uniformity of the helical structure deposition is investigated. Figure 1b inset shows the top view by scanning electron microscopy (SEM) for 200 nm h-WO$_x$ thin film deposition without 5 nm of SiO$_x$ thin film. To investigate the coverage of the helical shape structure, the shadow mask in the sizes of 15, 50, 100 μm in diameter was used during the depositing process. The cross-section of the via is shown with the helical structures, which demonstrates the good coverage with the GLAD method (Figure 1c). Noted the A-B is the cross-section of the via size with a diameter of 15 μm designed shadow mask.

Figure 2a shows the current–voltage (I–V) characteristics in h-WO$_x$-based devices of 50 nm, 100 nm, and 200 nm. The 100 DC cycles are tested with the SET compliance current limit (CCL) operation. The volatile switching behaviors are shown in helical-shaped WO$_x$-based devices. The on-current reaches SET CCL of 1 mA in the devices with a helical layer of 50 and 100 nm (blue and green curves), while the self-compliance is observed in the device of 200 nm helical layer (red curve). The selectivity (S.L.) is defined as the current at on-voltage (i.e., 1 V) divided by the current at off-voltage (i.e., 0.3 V). The device

with a higher selectivity represents superior performance in reducing the SPC noise and crosstalk. The results show the h-WO$_x$ selection devices with active layers of 50 nm and 100 nm require the external current clamping circuit, and the power consumption in which is higher than those in devices of 200 nm h-WO$_x$ layer.

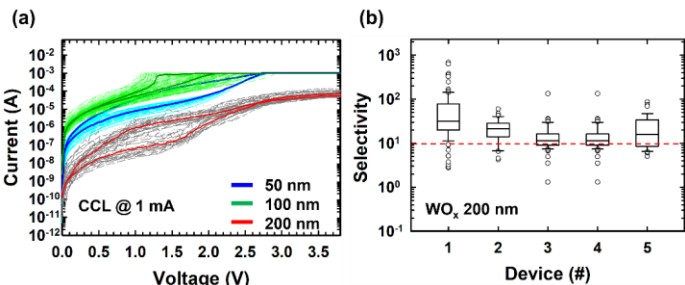

**Figure 2.** (**a**) I–V characteristics of volatile switching in WOx-based helical devices, (**b**) selectivity of WOx-based selection devices.

The off-current at the read voltage of 0.1 V is $8 \times 10^{-8}$, $2 \times 10^{-6}$, $1.4 \times 10^{-9}$ A for 50 nm, 100 nm, 200 nm devices, respectively. That is, the standby power is reduced with thicker h-WO$_x$ thin film, while the device is idle without operations of reading and writing. Although the drive-on voltage of h-WO$_x$ device at 1.52 V which is around three times of the transistor device in 1T-1R (e.g., 0.6 V), the power is reduced by two orders of magnitude. The power consumption is reduced 14 nW with h-WO$_x$ selection devices, as compared with 3 mW in 1T-1R configuration. Figure 2b shows the device-to-device (D2D) variation for the selection devices with h-WO$_x$ of 200 nm. The selectivity is in the range from 10 to $10^3$, and the median selectivity of 50 cycles for each device is above order of magnitude. Noted the selectivity larger than an order of magnitude to maximum of $10^3$ is sufficient for rectifying the non-volatile memory cell in the crossbar array. Future research on improving the selectivity [20,21] will continue by tailoring materials and device designs.

On the other hand, device reliability is the critical factor in evaluating the selection of devices. Figure 3a shows the one-time non-volatile switching after 100 DC cycling in four h-WO$_x$ (200 nm) selection devices at SET CCL of 1 mA, called the "pseudo SET process". The SET process with positive sweeping forms the conductive filament (i.e., low resistance state, LRS), and the RESET process with negative sweeping ruptures the filament (i.e., high resistance state, HRS) in a standard bipolar resistance random access memory (RRAM) device [22]. Herein, volatile switching is normally performed in devices before this pseudo SET process. That is, the starting current ranges from $10^{-8}$ to $10^{-10}$ A at 0.01 V. Notably, there is no forming process required in this helical-shaped device for volatile switching. That is, the volatile switching as shown in Figure 2a in the first cycle. The CCL of the electroforming process does not affect the device performance when it is below 1 mA. The forming CCL effect resulting in potential conductance quantization [23] will be discussed in our future work. In the memory crossbar array, the total due to leakage path from neighboring cells is still of micro-ampere scale which is the bottleneck the researchers tried to solve recently [24,25]. The volatile switching devices presented in this study show low current operation (<1 μA) using helical-shaped microstructures for future low-power memory applications.

With the DC cycling, the device is suggested under continuous bias stress and reaches SET CCL of 1 mA at around 3 V. Despite the pseudo SET process occurring, the vs. of the selection device is reconfigurable. Figure 3b shows the one-time refresh pseudo-RESET process, which shows a two-step conductance drop in the negative polarity sweeping process. The pseudo SET process occurs after the 100 cycles of vs. behaviors at positive sweeping (Figure 2a), the deep RESET process is performed once in negative polarity sweeps (i.e., dark red curve in negative polarity, Figure 3b). Then, the vs. behaviors have been recovered as the dark red curve in positive polarity in Figure 3b for the next 100 cycles as the selection device. The second "one-time refresh pseudo-RESET process brings the

vs. behaviors in h-WO$_x$ selection devices (i.e., light orange curve, Figure 3b). Noted the sweeping step of 10 mV was performed in the negative pseudo-RESET process from 0 to −2 V. Figure 3c shows the reconfigured volatile switching after the pseudo-RESET process in the identical four devices as in Figure 3a. In short, the volatile switching as the selection device using a helical-shaped active layer is low-power, recoverable and reconfigurable with an effortless pseudo RESET. This is thought to be suggested the confinement of the WO$_x$ helices which mitigates the vertical electrical field stress as compared with the continuous thin film devices. Future work will be performed to understand the physical mechanisms in discontinuous helical-shaped active layer for two-terminal vertical selection devices.

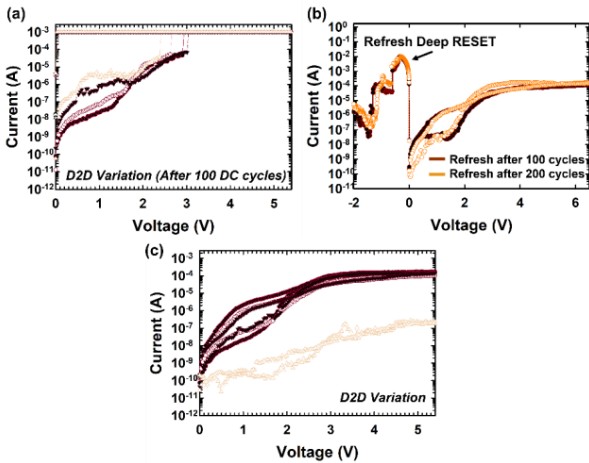

**Figure 3.** (**a**) Shorted one-time non-volatile switching (SET CCL: 1 mA), (**b**) refreshed one-time deep RESET cycle after 100 and 200 DC cycles, (**c**) device-to-device variation (median of 100 cycles).

The schematic diagram shows the possible mechanism of filamentary structure evolution during the volatile switching, cycling stressing, and the fresh RESET cycle on the helical-shaped microstructure (Figure 4). The helical structure is a non-continuous structure in 2-dimensional direction, and a continuous structure in 3-dimensional direction (i.e., spiral structure). The discontinuous oxygen vacancy distribution leads to the non-uniform filamentary formation (left panel) during the positive polarity DC sweep, where the SiO$_x$ thin film of 5 nm is as the oxygen reservoir. After a number of DC cycling which stresses the device with the repeating operation, the filamentary structure is suggested to be connected, i.e., LRS. Since the filamentary structure in the pitch between each helical-turn in cross section, the filament is not as robust as in WO$_x$ continuous helical-wire structure (blue). With refreshing RESET cycle, the filament ruptured at the pitch between each helical turn, and recovered the volatile switching based on the discontinuous filament. An investigation on materials and physical modeling will be the scope of future work.



**Figure 4.** Schematics for filamentary structure under operation conditions.

## 3. Conclusions

For the first time, a direct-grown helical-shaped h-WO$_x$ selection device with volatile switching behaviors for emerging memory applications is presented. The selectivity of volatile switching in the h-WO$_x$ selection devices is recoverable and reconfigurable with a one-time pseudo-RESET process on the negative voltage sweeping operation. The helical-shaped selection devices with the GLAD method are demonstrated with good compatibility, simple fabrication, low power, good selectivity, and good reliability for future embedded functional memory and security applications in the BEOL process.

**Author Contributions:** Conceptualization, Y.-C.C.; methodology, J.G.; validation, Y.-C.C., S.S. and Y.H.; formal analysis, Y.-C.C.; investigation, Y.-C.C.; resources, Y.H. and J.L.; data curation, S.S., Y.H. and Y.-C.C.; writing—original draft preparation, Y.-C.C.; writing—review and editing, Y.-C.C. and J.L.; visualization, Y.-C.C.; supervision, Y.-C.C. and J.G.; project administration, Y.-C.C.; funding acquisition, Y.-C.C. All authors have read and agreed to the published version of the manuscript.

**Funding:** This research received no external funding.

**Institutional Review Board Statement:** Not applicable.

**Informed Consent Statement:** Not applicable.

**Data Availability Statement:** Not applicable.

**Acknowledgments:** We acknowledge the support and consultation provided from Yao-Feng Chang (Intel Corporation) and Chao-Cheng Lin (Taiwan Semiconductor Research Institute).

**Conflicts of Interest:** The authors declare no conflict of interest.

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
