# Peer review of "Direct-Grown Helical-Shaped Tungsten-Oxide-Based Devices with Reconfigurable Selectivity for Memory Applications"

_jlpea, doi:10.3390/jlpea12040055_

Round 1

Reviewer 1 Report

This manuscript is devoted to the study of selection devices with helical-shaped WOx nanostructures. Interesting approach is proposed to fabricate such devices and some new results is presented and discussed. But, there are some comments for authors which should be answered before manuscript publication.

1. Observed drops on Figure 3(b) is an interesting fact, but it is practically don't discussed in text. These steps can be caused by conductance quantization. It would be useful to analyze them in terms of well-known approaches taking into account the effect of current compliance (see, for example, doi.org/10.1016/j.spmi.2018.07.006).

2. The discussion about the role of the helicon-shaped nanostructure on the device parameters should be expanded.

3. A large number of self-citations in the reference list (more than 50%). 

This manuscript could be published after the correction of this comments.

Reviewer 2 Report

I have read manuscript entitled “Direct grown helical-shaped tungsten oxide-based devices with reconfigurable selectivity for memory applications” in a fairly detailed fashion. This paper deals with the tungsten oxide helical-shaped device with volatile switching behaviors for potential application in emerging memory devices. In general, the topic of this manuscript is academically and technologically relevant. However, there exist some points that need to be clarified.

Information should be provided on the experimental section about the fabrication process of the tungsten oxide or to specify exactly which commercial tungsten oxide was used for fabrication devices under study in this manuscript.

The device characterization procedure is not explained. How was the excitation voltage generated, what form was the applied voltage?

What are the resistance values of the high resistance state ROFF and the high resistance state RON? From the given graphs, it seems that the ratio ROFF/RON is small, which is an unfavorable characteristic of fabricated devices.

It is necessary to examine the change in memristance (resistance) as a function of the number of the applied voltage pulses.

Also, it is necessary to examine the influence of temperature on stability.

On the basis of which facts, the authors base their claim that the manufactured device is compatible with CMOS technology? In general, the conclusions in this manuscript are not strongly supported by the experimental results

A careful revision of English is also recommended.

Based on the above, I think that manuscript in this version cannot be accepted for publication in Journal of Low Power Electronics and Applications since it needs a major revision.

Reviewer 3 Report

The work reports a WOx-based volatile resistive switching device. The authors claim that this device can be used as a reconfigurable selector in crossbar array. However, the device performance is far from the performances of reported devices.

1.     Much higher selectivity (10^5~10^7) has been extensively reported in threshold switching devices (for instance, Adv. Mater. 30, 1802516,2018; Jiang, H. et al. Nat. Commun. 8, 882, 2017), OVS devices (), etc.

2.     The under-through current of selector devices should be generally higher than the currents of the memory devices for selection. The highest on/off ratio (~10^3) can only be achieved at quite low current ranges (<1uA at 1V, red curves in Figure 2), which makes the reported devices having limited uses in selector applications.

Reviewer 4 Report

Memristor and memristive systems are the basis of new paradigm of electronics leading to the creation of novel information-computing systems. The present paper contributes well to this research trend by demonstrating original selector device based on volatile switching in helical-shaped tungsten oxide grown by the glancing angle deposition method. Good selectivity values, low power consumption and reliability of the proposed device are important to overcome the sneak path current problem of memristive crossbar arrays. The results are of immediate interest for memristive community and relevant to the scope of the JLPEA journal. The paper can be acceptable for publication after minor revision in response to the following comments.

1) The Authors should explain in a schematic diagram the possible mechanism of filamentary switching taking into account all the deposited layers and complex morphology of the helical WOx structure.

2) Self-citation percent should be significantly decreased.

Round 2

Reviewer 2 Report

I suggest that the manuscript be accepted in its current form.

Author Response

Dear Editor and Reviewers,

Please see attached files as the response Letter and revised manuscript for Manuscript ID jlpea-1892809 entitled "Direct Grown helical-shaped Tungsten Oxide-based Devices with Reconfigurable Selectivity for Memory Applications" which submitted to Journal of Low Power Electronics and Applications (JLPEA).

Reviewer’s comments:

I suggest that the manuscript be accepted in its current form.

Thanks reviewer’s positive feedback and recommendation to publish the manuscript.

Editor’s comments:

This is an interesting report. Upon perusal of the revised manuscript, the authors have improved the work quality with substantial revisions and modifications. The questions and concerns have been addressed satisfactorily in most. With sufficient novelty, high quality, and potential significance, it warrants publication in JLPEA upon below minor revisions.

Thanks to the editor’s consideration and positive feedback on this manuscript.

In the Introduction part, although the authors mentioned the necessity of mitigating the development gap between 1T-1R and self-rectified memory, the novel two-terminal selection device is proposed and demonstrated with simplified fabrication, low operation energy, and scalability. But, there is a lack of specific latest advances and progress in this field. Authors are encouraged to cite the highly relevant references in the final version of the revised manuscript to enhance the introduction part's completeness and readability.

Thanks to editors’ valuable comments.  We’ve revised the manuscript and added the relevant references for the introduction section to describe the current obstacles in the device-level development for the low power memory technology. The circuit- and system-level studies on latency and energy consumption for 1T-1R or multiple-T-1R configurations have been reported. However, there is a pressing need for the device-level solution towards the SPC-induced operating error and power consumption issues. 

We’ve revised the manuscript accordingly. We’ve also reduced the self-citation to under 10% of references on the manuscript.

Recently, self-rectified memory applications without the integration of the selection devices were published. However, the filamentary-based memory has intrinsic high off-current (~ 10-6) after electroforming which leads to the bottleneck of energy efficiency as well as noise immunity in the current self-rectified memory [9-12]. To mitigate the development gap between 1T-1R and self-rectified memory, the novel two-terminal selection device is proposed and demonstrated with simplified fabrication, low operation energy, and scalability. To investigate the system-level latency and writing energy, the SPICE modeling on 1T-1R, dynamic random-access memory (DRAM), and resistive random access memory (RRAM) (~1 Gb) have been reported [13]. The 2T-1R and 3T-1R configurations are also published to mitigate the SPC-induced operation errors while reducing the energy consumption [14-17]. Despite there are breakthroughs on circuit- and system-level studies, the device-level RRAM characteristics still lack investigation. To overcome the limitations on device-level memory development, the power consumption and SPC-induced errors in the array operation need an urgent solution. In this work, the direct-grown two-terminal selection device that uses the nano helical structures is presented with reconfigurable selectivity. The results present the reconfigurable volatile switching (VS) behaviors of WOx helical-shaped films, which is suggested as the solution for the SPC issue without compromising the structure simplicity, which enables the low power, high density, CMOS compatible technology towards the crossbar memory array applications.
